# A Review of the Removal of Dyestuffs from Effluents onto Biochar

**Prakash Parthasarathy** [1,*], **Samra Sajjad** [2], **Junaid Saleem** [1], **Mohammad Alherbawi** [1] and **Gordon Mckay** [1]

1 Division of Sustainable Development, College of Science and Engineering, Hamad Bin Khalifa University, Qatar Foundation, Doha 5825, Qatar; jsaleem@hbku.edu.qa (J.S.); mnalherbawi@hbku.edu.qa (M.A.); gmckay@hbku.edu.qa (G.M.)

2 Centre for Advanced Materials, Qatar University, Doha 2713, Qatar; samra.sajjad@qu.edu.qa

* Correspondence: pparthasarathy@hbku.edu.qa

**Abstract:** The study provides a review of various applications of biomass-derived biochars, waste-derived biochars, and modified biochars as adsorbent materials for removing dyestuff from process effluents. Processing significant amounts of dye effluent discharges into receiving waters can supply major benefits to countries which are affected by the water crisis and anticipated future stress in many areas in the world. When compared to most conventional adsorbents, biochars can provide an economically attractive solution. In comparison to many other textile effluent treatment processes, adsorption technology provides an economic, easily managed, and highly effective treatment option. Several tabulated data values are provided that summarize the main characteristics of various biochar adsorbents according to their ability to remove dyestuffs from wastewaters.

**Keywords:** effluents; dye removal; biochar; adsorption; dye absorption capabilities

## 1. Introduction

Dyestuffs color and pollute receiving waters, streams, and rivers as a result of inadequate processing of the industrial effluents by a variety of industrial applications including the food and beverage companies, paper and pulp processing, paint manufacturing, pharmaceutical processing, printing, textiles, dyeing, and printing [1]. Many dyes pose a grave danger to the water environmental ecosystem due to their chemical properties, with serious consequences for human health, animal, and plant ecosystems [2,3]. Aside from a limited number of studies indicating that specific dyes are toxic, the presence of dyestuffs into receiving waters reduces the photosynthetic process by inhibiting light from passing through [4]. During the degradation process, dyes consume the dissolved oxygen concentrations of the receiving water, therefore decreasing the water quality standards for aquatic species. This has detrimental visual aesthetic impacts which may result in health reproductive issues in fishes [5]. Specific dyes have a negative impact on the skin, kidneys, liver, reproductive system, heart, brain, and nervous system, and some may be carcinogenic or mutagenic.

Data on dyestuff effluent discharge volumes and production quantities are not readily available or recorded around the world. According to available data, 700,100 tons of dyestuffs are produced every year for 10,000 dyes. According to industry figures, the global dyestuffs produced yearly is 1.8 to $1.9 \times 10^6$ tons with more than 11,000 dye pigments applied primarily in the food, textile, cosmetics, leather, paper, and plastics industries [6]. Depending on the type of dyestuff and the process technology used, 1–10% of dye is not used in the dyeing process, indicating that significant amounts of dye are discharged to the water bodies via various means [7].

The majority of dyestuffs have specific characteristics such as chemically stable and light fastness [8]. Furthermore, the dye color reduces light penetration in streams and rivers, therefore reducing photosynthesis and dissolved oxygen content. They prevent a

variety of chemical functions based on the material to which they are applied and the color they impart (Figure 1). All of these properties are advantageous to the dye user and are enhanced by dyestuff manufacturers. However, the huge volumes of effluent makes the treatment of these dyes to comply with environmental effluent discharge standards very problematic. In addition, water can be colored in certain cases with dye concentrations as little as 1 ppm. The majority of dyeing applications employ copious amount of water during the dyeing, washing, and rinsing stages [9,10].

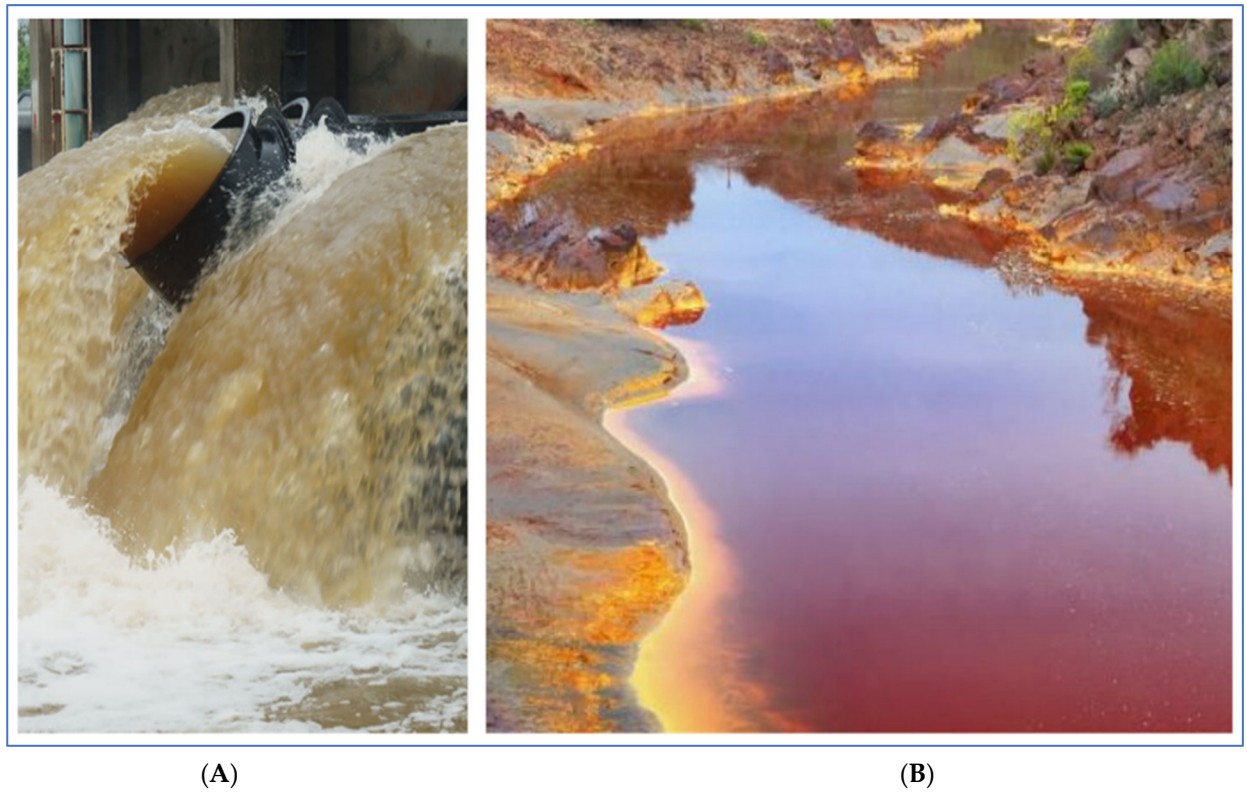

**(A)**　　　　　　　　　　　　　　　　　　　　　　　　　　　　　　　　**(B)**

**Figure 1.** (**A**) Dye house discharge. (**B**) River quality affected by dyestuffs.

The majority of recent review publications have focused on the biochar manufacturing process and dye removal by adsorption. However, there is a lack of data on the classification of dyes and their properties, as well as the removal of such dyes using biochar as an adsorbent material. This review examines the classification and qualities of dyestuffs in this setting. This article provides a rapid overview of various wastewater treatment systems, followed by a detailed explanation of the benefits of using adsorption technology. The key properties and applications of standard and modified (altered) biochars for color removal are also highlighted in the review. The investigations on the research reported in the directions of biochar modification and application marked a step toward the practical application of biochar.

## 2. Dye Classifications

Dyes are colored molecules or ions that can be applied to a wide range of materials including food, beverages, and textiles in solution or as a dispersion. Most dyes have a high-water solubility, and often contain a sulfonic acid group, usually in the form of a sodium salt, which is responsible for the solubility of many water-soluble dyestuffs [11]. Dye colors are created by chemical groups absorbing light of various wavelengths in the visible region of the spectrum. Different unsaturated chemical groups on chromophores promote this key distinguishing property. Figure 2 depicts the more common ones.

**Figure 2.** Color-producing chromophores or groups.

Auxochromes are groups that can also enhance the water solubility and improve the dye absorption potential for adsorbing material; examples include substituted sulfonic, hydroxyl, carbonyl, or amino groups. Dyes can be classified based on their chemistry or their types of application. As a result, the chemical structure and type of dye must be a primary consideration in determining which dye wastewater process treatment technology should be applied for effluent removal, as well as determining what adsorbent properties are required for the adsorption of the specific dyestuff type.

*2.1. Reactive Dyes*

Reactive dyes are used extensively in the dyeing of cellulosic textile fibers, namely, flax and cotton. Due to their high adhesion to a substrate, they can also be used to dye linen, viscose, and silk [12,13]. These reactive compounds in the dye can form chemical bonds with textile uptake of fibers. The uptake of the dichlorotriazine type of reactive dye which becomes attached to the cellulose fiber by displacing the chloride grouping is depicted in the mechanistic schemes below. One or both chlorides may be present. Figures 3 and 4 show the typical dye uptake mechanisms for dyeing cellulosic materials.

**Figure 3.** Typical mechanisms for dyeing cellulose.

**Figure 4.** Reactive dyeing mechanism on cellulose.

Due to reactive dyes having a strong bonding affinity for cellulose, consequently, the hydroxyl group containing biosorbents have demonstrated a very strong dye uptake capacity to remove reactive dye compounds from textile dyehouse effluents [14].

### 2.2. Disperse Dyes

Disperse dyes are non-ionic substances that are commonly applied to polyesters but can be used in acetate or nylon fabrics. These dyes are water soluble and can be used for these fibers by diffusion into the fibers at increased temperatures. As there are no basic chemical groups, there are no attractive sites for acid dye groups, despite a weak attraction for basic dyes. The dye attachment mechanism is based on weak Van der Waals forces and dipole-dipole interactions, implying that like mechanisms may occur during the removal of disperse dyes onto biochar adsorbents [15]. Figure 5 depicts disperse blue 6 as an example of this class.

**Figure 5.** Disperse Blue 6 dye compound.

The dispersed dyestuff occurs typically as a fine suspension that can be filtered from the effluent discharge by biochar.

### 2.3. Vat Dyes

The majority of vat dyes have a ketonic style chromophore which may be applied to color cellulosic fibers and materials such as viscose, cotton, and linen. This is a broad category of dyes that includes indanthrones, anthraquinones, carbazoles, benzanthrones, polycyclic quinones, and acridones.

Figure 6 shows the structures of a typical vat dye [16].

**Figure 6.** Structure of the vat red 13 dye.

The large anthraquinone groups suggest that the removal process may involve electron clouds of the anthraquinone dye by adsorption onto the positively charged surface groups and adequately sized pore diameters of biochars.

*2.4. Direct Dyes*

Direct dyes or substantive dyes, as there is no fixation phase necessary, may be applied to color cotton yarn, viscose, and loose cotton of fabrics [17]. Mordant chemicals, such as chromium compounds that can undergo complexation by attaching substrate to chromophore to form an insoluble color, are used in some direct dyes, but not all, to fix the dye and improve color fastness. In the case of dark color shades such as black or navy blue dyes, this technique has proven to be cost-effective in achieving high color fastness. These dyes are now being reviewed due to environment and safety concerns which have limited their use. The mechanism for the application of direct dyes involves establishing non-ionic forces to attach the dyestuff to the textile fiber material [18]. The structure of direct yellow 24 dye is depicted in Figure 7.

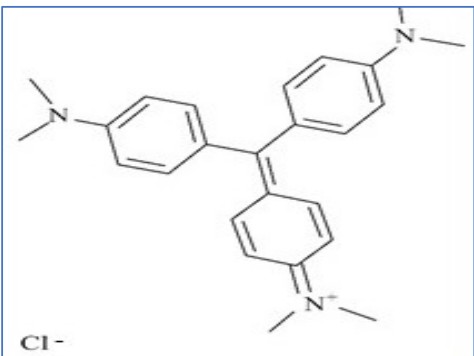

**Figure 7.** Structure of the direct yellow 24 dye.

Direct dyestuffs produce negatively charged ions in solution and can be adsorbed onto positive sites on biochars.

*2.5. Basic Dye*

Congo red (CR) is a member of the very large group of basic dyes that are characterized by the color, high tinctorial strength, and brilliance. These basic dyes are most commonly used on acrylic fibers, but they can also be used on other fibrous textiles when mordants are used. Furthermore, basic dyes are soluble media but not soluble in alkaline solutions. These dyes are primarily made up of imino or amino groups that are linked to triarylmethane or xanthene; they are also used in typewriter ribbon, carbon paper, and inks [19]. Monoazo, methane, and oxazine are the three main subclasses. Figure 8 depicts the structure of a basic CR dye.

**Figure 8.** Structure of basic crystal violet (CV) dye.

Since most basic dyes ionize in water, they have a positively charged colored ion or cationic species. These dyes are commonly classified as cationic dyes and adsorb most effectively onto negatively charged functional groups on the biochars [20].

### 2.6. Acid Dye

Acid dyestuffs possess a high color fastness and have a strong affinity for protein molecules, therefore, they are ideal for nylon, and wool, as well as silk. The majority of acidic dyestuffs are sodium salts based on sulfonic acid of organic species [21]. They are distinguished by various groupings of unsaturated aromatic rings known as the chromophore [22]. Concentrations of acidic dyes, even as low as 1 ppm, in the effluent raise the chemical oxygen demand (COD) level of the receiving waterbody and cause a visual disturbance to the environment [23]. Several dyes (i.e., arylamine-based types) are hazardous as they are carcinogenic or toxic [24]. The anthraquinone group is a major group of acid dyes. Figure 9 shows an example of the acid dye.

**Figure 9.** Structure of the acid blue 78 dye.

As the colored section is the anion of a sodium salt, acid dyes are frequently referred to as anionic dyes, possessing a colored ion based on sulfonic acid groups or, possibly, a carboxylic acid group which has the capability for adsorbing onto positive biochar sites.

### 2.7. Azo Dyes

Azo dyes account for over half the world's yearly dye production. They are broadly utilized in different industries, including food processing, pharmaceutical, leather, cosmetic, and textile dyeing applications. As shown in Figure 10, the azo dyes possess one or more azo functional groups (-N=N-) as the chromophoric species and are very frequently occurring in synthetic dyes with the aromatic conjugate ring structure [25]. When azo dyes are absorbed, they are metabolized by the intestinal microflora of the liver, resulting in the formation of aromatic amines which are a health hazard; furthermore, the azo dyes themselves, if water soluble, may become carcinogens.

**Figure 10.** Typical structure of chrysoidine azo dye.

Consequently, a number of countries have enacted legislation governing the manufacture, handling, and use of azo dyestuffs. The implemented legislation for azo dyes has been divided into two categories (i) those capable of producing metabolic carcinogens, and (ii) those not capable of producing carcinogenic species [26].

### 2.8. Sulfur Dyes

Sulfur dyes are widely produced and used in cotton-related industries because they are simple to apply, possess high color fastness, and are inexpensive. At room temperature, the dyes are insoluble in water; however, in an alkaline pH in the presence of a reducing agent and high temperature, these dyes become water soluble, allowing them to be adsorbed onto and thus dyeing the fabric. Environmentally, sulfur containing dyes are very polluting because of dye effluents and, as a result, are steadily losing favor [27]. Figure 11 depicts a typical sulfur dye.

**Figure 11.** Structure of the sulfur blue 7.

Due to there being an abundance of lone pairs of electrons due to oxygen, sulfur, and nitrogen, sulfur dyes can be adsorbed onto the positive surface sites, especially in the case of chelating groups acquiring lone electron pairs.

### 2.9. Aniline Dyes

The aniline class of synthetic dyes are predominantly made from coal tar that contains one or several phenyl groups. Around them, a new dye production industry was developed many years ago, manufacturing several of the most widely applied and well-known dyestuffs, such as malachite green (MG) and CV. The dye is found in the form of fine powders. Aniline dyestuffs are soluble in water and are produced in a variety of colors suitable for wood, leather, and fabric [28]. Some aniline dyestuffs have recently been linked to an increased cancer risk factor, according to research. Figure 12 depicts a typical aniline dye.

**Figure 12.** Structure of the mauveine A aniline dye.

As these dyestuffs are a cationic species [20], they can adsorb well on negative biochar groups. Additionally, electron clouds on aromatic ring species and nitrogen lone electron pairs are drawn to the positive sites on biochars.

### 2.10. Metal Complex Dyes

Metal complex dyes, also known as premetallized dyes, are insoluble in water. Such dyes are primarily monoazo compounds that are used to color wool, polyamides, silk, and nylon. The typical structure contains monoazo basic functional groups with some attached hydroxyl, amino, and carboxyl groups, giving these dye compounds a strong bond-forming capacity as well as the ability to produce coordination compounds with various transition metals such as cobalt, chromium, nickel, and copper. However, because their effluent discharges contain unfixed dye and their associated metal species, these metal complex dyes are now the cause of health, environmental, and safety concerns [15]. Figure 13 depicts the structure of a dithiolene metal complex dye.

**Figure 13.** Structure of the dithiolene metal complex dye.

These dyes can ionize in water, forming colored anionic species that may be adsorbed onto positive biochar sites; however, removing the toxic heavy metal counter ion, which requires negative sites on the biochar adsorbent is of greater concern.

### 2.11. Mordant Dyes

Most naturally occurring dyes cannot attach and fix strongly to many textile fibers and must be combined with a mordant compound that binds to both dye and the fiber. This can be used on both the fiber and the dye itself and it can be applied to both wood and nylon. This mordant substance creates a bond between the fiber and the dye molecule [29], however, other dyes have begun to replace mordants in recent years, in particular, direct dyes. The structure of mordant red 11 is depicted in Figure 14.

**Figure 14.** Structure of mordant red 11.

Mordant dyes ionize in solution, producing colored anionic species, necessitating adsorbing treatment with positive functional sites on the biochar.

## 3. Dye Removal Technologies

A variety of treatment processes are used in typical dyehouse effluent treatment technologies. These are generally classified into three types: chemical, biological, and physical processes. These technologies can be employed separately or collectively to achieve the best balance of multiple criteria including economic feasibility and process efficiency.

### 3.1. Physical Process Treatments

Adsorption [30] and biosorption [31] mechanisms describe the method of attraction of dyes onto the periphery of adsorbents in the form of particulates, granules or powders by physical bonding, strong chemical affinities, Van der Waals forces, or exchange of ions with surface functional sites on the adsorbent material surfaces via chemisorption or high energy chemical affinities.

To remove pollutants and colors from wastewater, flocculation and coagulation have long been utilized. This involves adding chemicals to dirty wastewater, often with agitation, and then permitting the pollutant-laden coagulants/flocculants to settle at the vessel bottom. Additional tertiary treatment or recycling may be applied to the refined water overflow that is discharged into water streams. The dye is adsorbed on the periphery/surface of the flocculant/coagulant without undergoing any chemical decomposition, similar to adsorption. Two typically utilized materials are ferrous sulfate and alum [32]. A high settling time for cleared water, the large amount of dye-laden flocculant/coagulant produced and requiring dumping, and the economics involved with the drying of highly wet slurry preceding landfill dumping are all disadvantages.

Electrokinetic coagulation and irradiation [33] are two other alternatives. The use of aluminum electrodes in electrocoagulation has been extensively studied. They unleash charged aluminum ions in solution [34], resulting in additional active or charged adsorption sites. However, they also have the drawbacks of coagulation. Catalytic irradiation/irradiation for removal of color has been researched in some circumstances. The efficacy of methylene blue (MB) removal was examined. The removal effectiveness for the maximum involvement of Pd catalyst reached around 86.4 percent after 120 min of irradiation [35], showing a five-fold improvement in removal rate. The two procedures were highly effective in removing some dyes but less so at removing others [36]. A more detailed examination into these procedures is required in order to verify them.

Membrane filtration is the process of physically separating soluble colors in water effluents using permeable membranes under pressure [37]. The techniques such as ultrafiltration and nanofiltration are also gaining popularity [38,39]. The aforementioned traditional pathways are analogous to membrane technology. Osmosis (forward and reverse) [39–41] and filtration (ultra/nano) [40,41] may both separate molecules larger than 1 nm. The high cost of pumping energy and the likelihood of membrane pore obstruction are two of membranes' major flaws. In case the dye effluents were previously employed for dyeing clothing/textiles, textile threads and fine particles are the ones that are commonly cleared along with the spent liquid. A pre-filtration stage probably is advantageous in these situations to prevent blockage and early membrane fouling [42]. If a condensed dye coat formed, the net process efficiency drops as the flow rate is reduced. Furthermore, the expense of filter media becomes increasingly significant as the unit's capacity increases. Several recent researches [43,44] have employed photocatalysis and ultrasonics to increase dye adsorption on solid particulates, particularly when magnetic biochars and nanoparticles were used. One drawback of utilizing photocatalytic processes to remove dye at full scale is the preparation of the photocatalysts, which can significantly change the cost of the process, whereas ultrasound treatment has the disadvantage of consuming very high amounts of energy.

### 3.2. Chemical Process Treatments

Ultraviolet light degradation has little effect on dyes. Most chemical techniques for the treatment of dye-polluted streams seek to knock down the intricate dye particles into simpler and less hazardous chemicals such as $CO_2$ or $H_2O$ through a sequence of oxidative reactions. For example, chlorine, wet air oxidation (WAO), ozonation, and other advanced oxidation processes (AOP).

Although WAO [45] is found to be efficient in the treatment of dye effluents, catalysts have been suggested as a way to improve process effectiveness [46,47]. Ozone is commonly employed in commercial AOP's for the elimination of organics and color [48]. Ozonation catalysts are being investigated [49]. Other powerful oxidizing reagent chemicals viz. sodium hypochlorite, chlorine, and hydrogen peroxide are being utilized commercially in the oxidation of dye-polluted streams [50,51]. These reagents break down dyes into simpler compounds [52]. Nonetheless, there are evidences that the benefits of utilizing chlorinated reagents may be exceeded by the production of harmful byproducts (such as chloroanilines) [53]. Oxidized byproducts and bromoforms are also of concern and are extensively used in the area of ozonation [54], and several studies have investigated their toxic characteristics [55].

For dye removal from wastewater effluents, several catalysts such as $TiO_2$ or Fenton's reagents have been used as an alternate to typical oxidizers in AOP [56]. Through the application of biochar as a support [57,58] and, more recently, co-catalytic nanoparticles/carbon dots [59,60] as co-catalysts, $TiO_2$-based photocatalysts can improve dye removal performances by enabling synergistic adsorption and degradation. The AOP technique has been shown to have a high dye removal efficiency [61]. However, one significant disadvantage is the intricacy of the resulting ferrous sludge [62]. Many AOP procedures are in place. It is possible to remove up to 90% of the dye; however, a suitable pH level balance and the sludge treatment process must be considered. When using typical titanium dioxide for removing colors, photochemical deterioration [63] is a continuous subject of attention, but it is limited by the technique. Electrochemical treatment was applied to commercial azo dye effluents [64].

### 3.3. Biological-Based Processes

Traditional aerobic techniques viz. activated sludge has proved successful in treating polluted streams, nonetheless, they are unsuccessful at removing colors. Dye compounds are frequently hydrophilic and have a low affinity for biomass, resulting in early application [65]. Dyes are frequently resistant to microbial breakdown due to their poisonous and stable molecular structures. Dye molecules are broken down into soluble organics in the absence of oxygen, then to $CH_4$ and $H_2S$ by a sequence of microbial processes. Although reasonable results for reactive, azo, and diazo dye decolorization were documented [66], the procedure had no effect on hazardous aromatic amines in the treated stream [67].

## 4. Adsorption Process Technologies

Adsorption is a popular process for purification purposes at large scales. Various contaminants in solution and in gaseous emission streams may be removed by being attracted to the surfaces of several solid phase materials, referred to as adsorbents. Adsorbents have been made from a variety of materials for a variety of uses, including water treatment, dye separation, indicators, desiccants, and catalysis. An effective adsorption system may be capable of removing all the pollutants, releasing a contaminant-free fluid.

Due to its simplicity, ease of operation, simple design, and ease of scaling up, the adsorption process is considered a better alternative in the treatment of dye from industrial effluents. It also has a high capacity and a favorable rate, and it is insensitive to harmful chemicals [68]. It can also help to resolve the challenges of high energy input (used in reverse osmosis and UV sterilization), which is a problem in many developing countries. Adsorption is preferable to photocatalytic and ultrasound treatment processes because photocatalyst preparation is expensive and ultrasound treatment consumes a lot of energy.

### 4.1. Adsorbent Properties and Adsorption Mechanism

Adsorbent materials with high selectivity and porosity are required. Activated carbon materials have demonstrated tremendous removal of organics such as dyes [30,69,70]. The diffusion of dye on the surface of adsorbents is either by physical phenomena or chemical means. The dye materials bind to the adsorbent periphery by hydrogen bonding or Van der Waals forces in physisorption mechanism [71]. In chemisorption, the dye ions or molecules form chemical bonds with certain surface functional sites or groups. Section 2 describes various dye functional sites or groups that can involve in multiple adsorptive bonds on different adsorbents. Section 5 of this article explains dye removal via adsorption mechanism using biochar materials.

### 4.2. Adsorbents

The choice of adsorbent is largely determined by the uptake potential of the adsorbent material for a particular adsorbate. Preferably, the adsorbent material shall meet the following criteria [70,72]:

i. Reasonably a decent surface area and pore volume;
ii. Appropriate pore size dissemination and pore network;
iii. Adsorbent functional groups of the surface charge and appropriate type;
iv. Surface functional group types and charge on colored dye ions/group;
v. pH of the solution that is appropriate for uptake.

As many dyestuffs are large molecules, it is critical to compare the dye particle dimensions to the pore (aperture) size distribution of biochar in the first two categories to make certain that the dye particles can penetrate across the pores easily. Furthermore, dye molecules frequently form larger groups or micelles by interacting with one another. As a result, mesoporous adsorbents may be preferable to those with a microporous design and a decent surface area. A high surface area, on the other hand, are usually beneficial. A broad mesoporous network benefits adsorption kinetics and process design by allowing for quicker diffusion.

Clay, silica, kraft lignin, fly ash, sludge, slag, and red mud are some of the most often utilized natural adsorbents. Activated carbon [73], activated alumina [74,75], silica gel, and zeolites are commercially available adsorbents [76]. Agricultural residues/wastes viz. sugarcane bagasse, rice brans, lignocellulosic biomasses, fruit stones, and nut hulls [77–80], inorganic materials viz. clay, Fuller's earth, bentonite, lignite coal, peat, chitosan, and io-exchange resin materials [81] are also used as adsorbents. New adsorbent materials such as CNTs, MCM-41 [82,83], and molecular sieves [82] have been discovered in recent literature.

As long as the appropriate adsorbent materials with decent adsorption potential are accessible at a reasonable price, adsorption is an attractive color removal method because it uses large amounts of water and has relatively low dye concentrations. Consequently, biochars have a good chance of being used to remove dyes from effluent. As opposed to activated carbons, biochar is generated through pyrolysis technique at temperatures between 623 and 873 K, and by employing inexpensive biomass materials. Numerous operational factors, including feedstock type and pyrolysis temperature, influence biochar attributes, resulting in products with such a broad range of specific surface area, pore volume, carbon content, volatile content, ash, pH, and cation exchange capacity (CEC). The formation of biochar with a highly evolved high porosity, specific surface area, pH, carbon, and ash content, but low CEC and volatile content, is aided by a high pyrolysis temperature, which is most closely attributable to a high level of breakdown of organic materials. Even at greater pyrolysis temperatures, biochars made from animal and solid wastes have lower carbon content, volatile content, CEC, and surface areas than biochars made from woody biomass and agricultural residues [84]. The reason for this discrepancy is because the content of lignin and cellulose, as well as the moisture content of biomass, varies greatly among plant biomass wastes [85]. During the preparation phase of biochars, modifying agents are also added to create highly charged surface properties. Several instances of altered and unmodified biochar materials are included in Section 5.

## 5. Adsorption of Dye onto Biochar Materials for Dye Removal

*5.1. Biochar as a Dye Removal Adsorbent*

Activated carbons are amorphous carbon-based substances with decent porosity and large internal surface area. Its feedstock can be almost any organic substances with a relatively decent carbon composition, covering traditional materials such as hard and softwood, coconut hull, peat and lignite coal, natural and artificial polymers. Surface areas for marketable carbons typically fall between the 500–1500 m$^2$/g range [86] and can even reach 3000 m$^2$/g. Carbon materials are again classified into two sub-categories based on whether they are used to remove pollutant particles from fluids.

The first ones are typically microporous with a pore size of 2 nm diameter and are usually granular, whilst the latter are mesoporous materials with a pore size ranging between 2–50 nm diameter and are usually in powder form [71]. Both types are useful in wastewater treatment, where they aid in decolorization, odor removal, metal recovery, and organics adsorption. The pore volume, internal surface area, and size distribution are all proportional to its adsorption capacity.

Organics have been reported to adsorb in pores just fitting the adsorbate molecule [45]. Humic acids and dyes with size ranging between 1.5 and 3.0 nm that support adsorption phenomenon in mesopores are examples [87]. As a result, the biochar's pore size dissemination influences its adsorption potential for ions of varying size and shape. The electric strength between the adsorbate and the adsorbent (carbon surface) has proven to improve dye removal efficacy greatly.

The dissociation equilibria and functionality of specific functional sites on biochar material surfaces, such as carboxylic-lactonic groups, phenolic-alcolohic hydroxyl groups, aromatic-heterocyclic carbons, ketone-carbonyl groups, pyridinic-N, pyrollic-N, and quaternary-N nitrogen species, can influence adsorption. These potential biochar surface sites are influenced by several factors, including biomass source, pyrolysis parameters such as heating rate, temperature, residence time, nature of pyrolysis, etc.

Despite that activated and modified carbons are broadly used as adsorbent materials, they are relatively expensive due to the high costs of raw materials, energy, and chemical production. As a result, many researchers have focused on developing novel, high-capacity, low-cost adsorbents obtained from biomass residues. Metal organic frameworks and nano-adsorbent substances have lately been used to create highly efficient adsorbents [88]; however, the cost of treatment renders these materials prohibitively expensive.

As a result, in recent years, several low-cost adsorbents known as biochars have been produced by biomass pyrolysis and used in polluted water treatment applications. The technique is affordable and cost-effective only when the adsorbent is inexpensive and copious [89]. Pyrolysis of biomass leftovers into value-added biochar materials is a cost-effective process that produces high-value-added products: syngas and bio-oil. The pyrolysis process requires energy to run, but the process is driven by the by-products of the side reactions, and biochars possess a larger surface area and pore volume, as well as chemically functional moiety content, making them a much more potent adsorbent material than the biomass feedstock [90–93].

There are over a thousand papers on color removal in the literature, with over a hundred of these based on dye elimination employing biochar substances, including biochar products derived from vermicompost, cabbage residues, algae, and animal litters [94–98]. There have also been numerous publications on the synthesis and usage of altered biochar materials for dye color elimination. The dye potentials of unmodified/unaltered biochars and others are shown in Tables 1 and 2.

Table 1 shows the adsorption properties for cationic dye uptake onto unmodified biochars. MG [99–103], MB [104–109], rhodamine B (Rh B) [99], basic red 9 (BR 9), and CV [103,110] values are included in the data. Many citations only present the quantity of dye removed (in %) [99,100,104,106,110,111], which is valuable, however, this value varies with adsorbent quantity, dye concentration, and adsorbate volume.

Table 1. Basic dyes (cationic) and their adsorption onto biochars.

| Dye | Biochar Feedstock | Pyrolysis Conditions | | | Pore Volume (cm³/g) | BET Surface Area (m²/g) | Adsorption Capacity (mg/g) or Dye Removal (%) | Isotherm Type | Kinetic Model | Parameters | | Mechanism | Reference |
|---|---|---|---|---|---|---|---|---|---|---|---|---|---|
| | | Temperature (K) | Heating Rate (K/min) | Time (min) | | | | | | pH | Equilibrium Time (min) | | |
| MB | Date palm fronds | 973 | - | 240 | 0.134 | 430 | 205 | - | - | 6 | 36 | - | [104] |
| MG | Tapioca peel | 1073 | 10 | 180 | - | - | 32% | Langmuir, Freundlich | Pseudo I-order, Pseudo-II order | 2–10 | 0–180 | - | [99] |
| Rh B | Tapioca peel | 1073 | 10 | 180 | - | - | 66% | Langmuir, Freundlich | Pseudo I-order, Pseudo-II order | 2–10 | 0–180 | - | [99] |
| MB | *Chlorella* sp. *microalgae* | MW heating (2450 MHz, 800 W) | - | - | - | 3 | 110 | Freundlich, Temkin | Pseudo I-order, Pseudo-II order, Elovich | 2–10 | 7200 | Boyd, Intraparticle diffusion | [105] |
| MG | Rice husk | 673–873 | - | 60 | - | - | 65 | Langmuir, Freundlich | Pseudo I-order, Pseudo-II order, Elovich | 2, 4, 6, 8 | 1440 | - | [100] |
| MG | Crab shell | 1073 | - | 120 | 0.086 | 82 | 12,500 | Langmuir | Pseudo-II order | 7 | 2 | Electrostatic attraction, Hydrogen bonding, π-π interactions | [101] |
| MB | Areca leaf | 473 | 5 | 60 | - | 21 | 120 | Langmuir, Freundlich | Pseudo I-order, Pseudo-II order | 7 | 720 | Electrostatic attraction | [106] |
| MB | *Wodyetia Bifurcate* | 973 | 10 | 30 | - | - | 150 | Sips | Pseudo I-order, Pseudo-II order | - | 30 | - | [107] |
| MG | Waste wheat straw/wheat bran | 1073 | 15 | 90 | - | - | 1740 | Langmuir | Pseudo-II order | 2, 4, 6, 8, 10 | - | Electrostatic interaction, Chemisorption | [102] |
| CV | Waste wheat straw/wheat bran | 1073 | 15 | 90 | - | - | 175 | Langmuir | Pseudo-II order | 2, 4, 6, 8, 10 | - | Electrostatic interaction, Chemisorption | [102] |
| MB | Switchgrass | 873 | - | 60 | 0.029 | 255 | 40 | Langmuir | Pseudo-II order | 6 | - | Intraparticle diffusion | [108] |

**Table 1.** *Cont.*

| Dye | Biochar Feedstock | Pyrolysis Conditions | | | Pore Volume (cm³/g) | BET Surface Area (m²/g) | Adsorption Capacity (mg/g) or Dye Removal (%) | Isotherm Type | Kinetic Model | Parameters | | Mechanism | Reference |
|---|---|---|---|---|---|---|---|---|---|---|---|---|---|
| | | Temperature (K) | Heating Rate (K/min) | Time (min) | | | | | | pH | Equilibrium Time (min) | | |
| MB | Switchgrass- | 1173 | - | 60 | 0.058 | 640 | 200 | Langmuir | Pseudo-II order | 6 | - | Intraparticle diffusion | [108] |
| CV | Mango leaves | 1073 | - | 60 | - | 170 | 180 | | - | 8 | 48 | - | [110] |
| MG | *Ulothrix zonata* algae | 1073 | 15 | 90 | - | 130 | 5300 | Freundlich | Pseudo-II order | 2, 4, 6, 10 | 840 | Chemisorption | [103] |
| CV | *Ulothrix zonata* algae | 1073 | 15 | 90 | - | 130 | 1220 | Freundlich | Pseudo-II order | 2, 4, 6, 10 | 840 | Chemisorption | [103] |
| BR 9 | Bovine bones | 1073 | 10 | 60 | 0.271 | 90 | 50 | Langmuir, Freundlich | Pseudo-II order | 7 | 180 | - | [111] |
| BR 9 | Bovine bones | 1073 | 10 | 180 | 0.193 | 95 | 50 | | Pseudo I-order | 7 | 180 | - | [111] |
| MB | Sugarcane bagasse | 773 | 10 | 90 | - | 260 | 70 | Langmuir, Freundlich | Pseudo I-order, Pseudo-II order | 7.4 | 180 | Intraparticle diffusion | [109] |

MB adsorption capacities (Table 1) are 110, 150, 38, and 195 mg/g for biochar materials derived from microalgae [105], Wodyetia [107], and switchgrass at 873 and 1173 K [45,108], respectively. Pyrolysis temperature significantly influences the adsorption potential of switchgrass biochar. MB adsorption capacity values in the literature for sugarcane bagasse [109], phosphoric acid-treated olive seed carbon [112], and bamboo cane active carbon [113] are 110, 135, and 455 mg/g, respectively. Biochar adsorption capacities for MG dye are remarkably decent, with values of 12,500 mg/g on crab shell [101], 1740 mg/g on wheat/bran straw-fed larvae biochar [102], and 5300 mg/g on biochar synthesized from *Ulothrix algae* [103].

The adsorbent potential values reported in the literature for activated carbon made from grape processing residue [114], shrimp shell [115], and plastic waste [116] are 665, 320, and 1430 mg/g, respectively. The amount of Rh B adsorbed on the surface of biochar sourced from tapioca shell is 33 mg/g [99] compared to reported values of 77 using Acacia mangium wood-derived carbon [117] and 30–40 mg/g on activated carbons from carnauba, macauba, and pine nut wastes activated using calcium chloride and phosphoric acid [118].

On biochar from mango leaves [102] and *Ulothrix zonata* algae [103], two capacities for the removal of CV are listed: 175 mg/g and a quite high value of 1220 mg/g. The capacity values reported in the literature range from 600 mg/g for a bentonite-alginate composite [119] to 75 mg/g for chitosan hydrogel beads [120]. Table 1 shows the final values for the adsorption of BR9 onto biochar materials from animal residue [111] after 1 and 3 h of heat treatment. At one (90 m$^2$/g) and three hours (95 m$^2$/g), the BR9 capacities were 50 and 52 mg/g, respectively. The general trend in surface areas and pyrolysis times was followed by these capacities. Literature values are slightly lower but of relative magnitude, for example, 29 and 15 mg/g for sepiolite [121] and fish bone [122], respectively.

Table 2 shows the anionic dye adsorption properties on unmodified biochars. Acid orange 7 (AO 7) [104], CR [96,97,101–103,105,108,123–126], reactive red RR 120 [127], Remazol violet 5R (RV5R), Remazol orange 3R (RO 3R), Remazol blue R (RBR) [128], orange G (OG) [108], and methyl orange (MO) [129] are some of them.

Table 2 shows only the dye removal composition (%) for the adsorption of AO7 using biochar derived from groundnut shell. Only a few instances of AO7 adsorption capacity potential are documented in articles, and they range from 50 to 180 mg/g on fly ash [130], oxihumolite [131], and chemically reactivated sawdust [132].

As CR is one of the most researched anionic dyestuffs, the citations in Table 2 are merely illustrative. Adsorption capacity potential of biochars derived from *chlorella* microalgae species [105], *phoenix dactylifera* [125], cotton stalk [126], orange skin [123], carapace (crab shell) [101], activated carbon [124], *spirulina* algae species [96], wheat bran larvae [102], switchgrass (charred at 873 K and 1173 K) [108], and Ulothrix algae species [103] are 160, 25, 250, 90, 20,315, 230, 85, 8, 23, and 345 mg/g. The study on switchgrass indicated that at elevated pyrolysis temperatures, a high-quality biochar is generated.

Most investigations have found that the CR dye adsorption capacity potential is below 100 mg/g. The maximum value was observed from pyrolyzed crab shell with 80 m$^2$/g surface area. This result was achieved at a pH of 4 and volume to mass ratio of 2; nonetheless, at a CR concentration above 20 g/L. The activated carbon obtained from date stone exhibited a low adsorption capacity potential of 35 mg/g. The huge dye molecular size (695 g/mol) and the total pore volume of 0.086 are responsible for the poor capacities. CR dye, unlike other anionic acid dyes, is a direct dye with no anionic bonding characteristics [133,134]. The significance of examining adsorbent pore size distribution is shown by this phenomenon [135]. Microwave treatment was used following phosphoric acid activation to synthesize mesoporous activated carbon. The carbon showed a higher surface area and a 350 mg/g adsorption capability.

**Table 2.** Acid dyes (anionic) and their adsorption onto biochars.

| Dye | Biochar Feedstock | Pyrolysis Conditions | | | Pore Volume (cm³/g) | BET Surface Area (m²/g) | Adsorption Capacity (mg/g) or Dye Removal (%) | Isotherm Type | Kinetic Model | Adsorbent Parameters | | Mechanism | Reference |
|---|---|---|---|---|---|---|---|---|---|---|---|---|---|
| | | Temperature (K) | Heating Rate (K/min) | Time (min) | | | | | | pH | Equilibrium Time (min) | | |
| CR | *Chlorella* sp. microalgal | MW heating (2450 MHz, 800 W) | - | - | - | 3 | 160 | Langmuir, Freundlich, Temkin | Pseudo I-order, Pseudo II-order, Elovich | 2–10 | 240 | Boyd, Intraparticle diffusion | [105] |
| CR | Rice husk | 773 | 5 | 180 | - | - | 66–97% | Langmuir, Freundlich | - | 2, 4, 6, 7, 9, 11 | 5760 | - | [97] |
| RR 120 | *Eucheuma spinosum* | 573–873 | 10 | 120 | - | - | 330 | Langmuir, Freundlich, Temkin | Pseudo I-order, Pseudo II-order, Elovich | 3–9 | 20 | Electrostatic interaction, Ion exchange, Metal complexation, Hydrogen bonding | [127] |
| CR | *Phoenix dactylifera* leaves | 673 | - | - | - | 1 | 25 | Langmuir, Freundlich | Pseudo I-order, Pseudo II-order | 5.8 | 120 | - | [125] |
| CR | Cotton stalks | 673 | 8 | 90 | - | - | 250 | Langmuir, Freundlich, Temkin, Dubinin-Radushkevich | Pseudo I-order, Pseudo II-order | 2–10 | 180 | Electrostatic attraction | [126] |
| CR | Orange peel | 1073 | 15 | 15 | - | | 20 | | - | - | - | - | - | [123] |
| Remazol BV 5R | Green marine algae (*Caulerpa scalpelliformis*) | 573–773 | 5 | 120 | - | - | 70% | Langmuir, Freundlich, Sips, T | Pseudo I-order, Pseudo II-order | 2–5 | - | - | [128] |
| Remazol BO 3R | Green marine algae (*Caulerpa scalpelliformis*) | 573–773 | 5 | 120 | - | - | 77% | Langmuir, Freundlich, Sips, Temkin | Pseudo I-order, Pseudo II-order | 2–5 | - | - | [128] |
| Remazol BO 3R | Green marine algae (*Caulerpa scalpelliformis*) | 573–773 | 5 | 120 | - | - | 75% | Langmuir, Freundlich, Sips, Temkin | Pseudo I-order, Pseudo II-order | 2–5 | - | - | [128] |
| Remazol BO 3R | Crab shell | 1073 | - | 120 | 0.086 | 82 | 20,315 | Langmuir | Pseudo I-order, Pseudo II-order | 4 | 2 | Electrostatic attraction, Hydrogen bonding, π-π interactions | [101] |
| CR | Activated Carbon | 723 | 20 | 120 | - | - | 230 | Freundlich | - | 2–10 | 120 | - | [124] |
| CR | Spirulina platensis algae | 723 | 20 | 120 | - | - | | Freundlich | - | 2–10 | 120 | - | [96] |
| CR | Waste wheat straw/wheat bran | 1073 | 15 | 90 | - | - | 90 | Langmuir | Pseudo II-order | 2, 4, 6, 8, 10 | - | Chemisorption, Electrostatic interaction | [102] |
| OG | Switchgrass | 873 | - | 60 | 0.029 | 255 | 8 | Langmuir | Pseudo II-order | 6 | - | Outer boundary | [108] |
| CR | Switchgrass | 873 | - | 60 | 0.029 | 255 | 8 | Langmuir | Pseudo II-order | 6 | - | Outer boundary | [108] |
| CR | Switchgrass | 1173 | | 60 | 0.058 | 640 | 20 | Langmuir | Pseudo II-order | 6 | - | Outer boundary | [108] |
| CR | *Ulothrix zonata* algae | 1073 | 15 | 90 | - | 130 | 345 | Freundlich | Pseudo II-order | 2, 4, 6, 10 | 840 | Chemisorption | [103] |
| MO | Corn cob | 873 | 15 | 120 | - | 470 | 90 | Freundlich | Pseudo II-order | 5.6 | - | Physiochemical | [129] |

At temperatures between 513 to 553 K and reaction durations varying from 0.5 to 6.0 h, many bamboo biochars were generated [136]. Their CR absorption capabilities ranged from 30–100 mg/g, with the maximum values seen in biochars generated at high temperatures of 513 K and 553 K, as well as the longer treatment times of 5 and 6 h. After only 4 h at 523 K, an activated carbon obtained from apricot seeds [134] had a low CR capacity of 33 mg/g. This was ascribed to the small pyrolysis temperature and specific surface area. The latter two investigations' adsorption capacity potential was similar to that of date seed carbon. More research into the production of biochars through microwave and plasma pyrolysis techniques should be conducted.

Anionic reactive red (RR) 120 was adsorbed at a high (330 mg/g) capacity on biochar synthesized from Eucheuma spinosum [127]. $Fe_3O_4$-activated magnetic nanoparticles [137] and activated carbon [138] have high capacity values in the literature, exhibiting adsorption capacity potentials of 165 and 255 mg/g, respectively. Biochar made from green sea algae [128] has been reported to remove brilliant violet 5R, Remazol, and brilliant orange 3R dyes with removal percentages of more than 70%. On coffee shell activated carbon [139] and calcined eggshell [140], the reported figures for Remazol dyes are relatively low, at 65 and 15 mg/g for brilliant orange 3R and brilliant violet 5R, respectively.

Table 2 demonstrates that biochar derived from switchgrass [108] generated at 873 K displayed a poor orange G adsorption capacity of 8 mg/g. The poor surface area (255 m$^2$/g) of the char could have contributed to this low capacity. Other published values include 9 mg/g for activated carbon derived from Thespesia populnea [141] and 19 mg/g for nanoporous activated carbon [142]. All of these values indicate that orange G dye is one to be treated. When it came to adsorbing MO, corn cob char [129] had an adsorption capacity potential of 85 mg/g, while amidoxime char [143] had a potential of 140 mg/g.

*5.2. Dye Removal Using Adsorption onto Modified Biochars*

Several investigations are now being conducted to improve the adsorption efficacy of biochars. Many papers on various biochar modification techniques are available. Treatment or activation of biochar with bases and acids, chemical impregnation, size alteration, and encapsulation are some of the modification techniques. The cationic dye adsorption characteristics and performance properties are shown in Table 3, whereas the anionic dye adsorption characteristics and performance properties are shown in Table 4.

**Table 3.** Basic dyes (cationic) and their adsorption onto modified (altered) biochars.

| Dye | Modified Biochar Feedstock | Pyrolysis Conditions | | | Pore Volume (cm³/g) | BET Surface Area (m²/g) | Adsorption Capacity (mg/g) or Dye Removal (%) | Isotherm Type | Kinetic Model | Adsorbent Parameters | | Mechanism | Reference |
|---|---|---|---|---|---|---|---|---|---|---|---|---|---|
| | | Temperature (K) | Heating Rate (K/min) | Time | | | | | | pH | Equilibrium Time (min) | | |
| MB | Date palm fronds | 1073 | 20 | 240 | - | 70 | 210 | - | - | 7 | 180 | - | [144] |
| MG | Tapioca peel + S- doped | 1073 | 10 | 180 | - | 145 | 30 | Langmuir, Freundlich | Pseudo I-order, Pseudo II-order | 2–10 | 1080 | - | [99] |
| Rh B | Tapioca peel + S- doped | 1073 | 10 | 180 | - | 145 | 30 | Langmuir, Freundlich | Pseudo I-order, Pseudo II-order | 2–10 | 1080 | - | [99] |
| MB | Areca leaf + $K_2FeO_4^-$ | 473 | 5 | 60 | - | 20 | 250 | Langmuir, Freundlich | Pseudo I-order, Pseudo II-order | 7 | 720 | Electrostatic attraction | [106] |
| MG | Chitosan-tapioca peel + S-doped | 873 | - | 120 | - | 120 | 50 | Langmuir, Freundlich | Pseudo I-order, Pseudo II-order | 2–12 | 160 | Electrostatic attraction, Hydrogen bonding | [145] |
| Rh B | Chitosan-tapioca peel + S-doped | 873 | - | 120 | - | 120 | 40 | Langmuir, Freundlich | Pseudo I-order, Pseudo II-order | 2–12 | 160 | Electrostatic attraction, Hydrogen bonding, π-π interactions | [145] |
| MB | Sugarcane bagasse + steam | 1073 | 10 | 120 | 0.356 | 570 | 5220 | Langmuir, Freundlich | - | 7.4 | 180 | - | [146] |
| MB | Date palm fronds with Fe/Mn | 973 | 3 | 240 | - | 430 | 300 | Langmuir, Freundlich | Pseudo I-order, Pseudo II-order, Intraparticle diffusion, Elovich | 4–10 | 240 | Surface adsorption, π-π interactions, Ion exchange, Pore-filling | [147] |
| MB | Wakame *Undaria pinnatifida* leaves with calcination | 1073 | 10 | 120 | - | 1160 | 840 | Langmuir, Freundlich | Pseudo I-order, Pseudo II-order | 2–12 | 300 | Surface adsorption, Hydrogen bonding, π-π interactions, Pore-filling | [148] |

**Table 3.** *Cont.*

| Dye | Modified Biochar Feedstock | Pyrolysis Conditions | | | Pore Volume (cm³/g) | BET Surface Area (m²/g) | Adsorption Capacity (mg/g) or Dye Removal (%) | Isotherm Type | Kinetic Model | Adsorbent Parameters | | Mechanism | Reference |
|---|---|---|---|---|---|---|---|---|---|---|---|---|---|
| | | Temperature (K) | Heating Rate (K/min) | Time | | | | | | pH | Equilibrium Time (min) | | |
| Rh B | Wakame *Undaria pinnatifida* leaves with calcination | 1073 | 10 | 120 | - | 1160 | 530 | Langmuir, Freundlich | Pseudo I-order, Pseudo II-order | 2–12 | 300 | Surface adsorption, Hydrogen bonding, π-π interactions, Pore-filling | [148] |
| MG | Wakame *Undaria pinnatifida* leaves with calcination | 1073 | 10 | 120 | - | 1160 | 4065 | Langmuir, Freundlich | Pseudo I-order, Pseudo II-order | 2–12 | 300 | Surface adsorption, Hydrogen bonding, π-π interactions, Pore-filling | [148] |
| MG | Corn straw | 773 | - | 180 | - | 35 | 520 | Langmuir, Freundlich, Temkin | Pseudo I-order, Pseudo II-order, Intra diffusion | 2–9 | 20 | | [76] |
| MG | Rice husk + Cu + Al | 353 | - | 60 | 0.350 | 200 | 470 | Langmuir, Freundlich | | 9 | 200 | Pore-filling, π- π interactions | [149] |
| MG | Litchi peel + HC | 1123 | | 60 | 0.588 | 1010 | 2470 | Freundlich | Elovich | 8 | 720 | Hydrogen bonding, π-π interactions, Pore-filling, Electrostatic interaction | [150] |
| MG | Sugarcane bagasse + ZnCl₂ | 1073 | - | 120 | 0.0235 | 50 | 90 | Freundlich | Pseudo II-order | 8 | - | Boyd | [124] |

On Fe/Mn impregnated fronds pyrolyzed at 973 K [147] and 1073 K thermally treated fronds [144], the biochar from date palm frond [104] exhibited an MB adsorption capacity of 205 mg/g, whereas the biochars obtained from modified date frond demonstrated capacities of 300 and 210 mg/g. Biochar made from tapioca skin has a 30% MG removal capacity and a 65% Rh B elimination potential [99]. The removal capacity of biochar made from sulfur-doped tapioca peel was 75 and 90%, respectively [99]. 30 and 33 mg/g were reported as the greatest adsorption potential.

The sulfur-doped tapioca skin was coated with chitosan at 873 K, which raised the dye absorption capabilities to 50 mg/g for MG and 40 mg/g for Rh B, respectively [145]. Unaltered areca plant biochar showed an MG capacity of 190 mg/g [106]. After activating with $K_2FeO_4$, the adsorption potential of biochar prepared from wakame increased to 250 mg/g. The biochar sourced from chlorella microalgae displayed an MG adsorption capacity of 110 mg/g and is one among many seaweed/algae species biochars with potentials lying between 25 and 130 mg/g. The adsorption capability of MG on biochar obtained from chlorella was close to 80 mg/g [151].

Only 10 mg/g capacity of Rh B was accepted by seaweed biochar. Biochar made from calcined wakame seaweed, on the other hand, has extraordinarily high adsorption capabilities for MG, Rh B, and MB, with 4065, 840, and 530 mg/g, respectively [148]. Biochar generated from rice husk had an MG adsorption capacity of 65 mg/g [100]; after alteration with Cu + Al [149], the potential elevated to 470 mg/g. Biswas et al. [109] found that biochar made from sugarcane bagasse had a potential of 70 mg/g, while biochars made from steam-activated bagasse and $ZnCl_2$-modified bagasse had capacities of 5220 and 90 mg/g, respectively [109].

Table 4 shows the anionic dye adsorption properties on a variety of modified biochars. Biochar was prepared from rubber seeds and treated with NaOH at 1073 K. The adsorption potential of CR rose from 225 mg/g to 460 mg/g after activation [152]. The *shorea robusta* leaf extract offered a removal potential of only 2 mg/g for CR with an exceedingly poor surface area of 1.5 $m^2$/g after heating at 573 K [153]. The addition of Ag nanoparticles to this biochar raised its capacity and surface area to 23 mg/g and 21 $m^2$/g, respectively [153].

A biochar made from magnetic food waste with a removal potential of 23% was produced for application in a Fenton-style wastewater treatment system [43]. The biochar adsorption potential increased by 33% after ultrasonic treatment. Finally, ultrasonics + $H_2O_2$ + biochar achieved a 97% removal in 3 h. Table 2 shows that dactylifera leaf biochar (without any modification) exhibited a CR adsorption potential of 25 mg/g before Mn activation, however, it increased to 120 mg/g after Mn activation [125]. Marine chlorella vulgaris algae were transformed to biochar at temperatures of 723, 823, and 923 K [154], yielding surface areas of 265, 350, and 150 $m^2$/g, respectively. This adsorbent's adsorption potential for the anionic RY-45 dye was 48 mg/g (experimental) and 58 mg/g at 823 K (Langmuir).

**Table 4.** Acid dyes (anionic) and their adsorption onto modified (altered) biochars.

| Dye | Biochar Modified Feedstock | Pyrolysis Conditions | | | Pore Volume (cm³/g) | BET Surface Area (m²/g) | Adsorption Capacity (mg/g) or Dye Removal (%) | Isotherm Type | Kinetic Model | Adsorbent Parameters | | Mechanism | Reference |
|---|---|---|---|---|---|---|---|---|---|---|---|---|---|
| | | Temperature (K) | Heating Rate (K/min) | Time (min) | | | | | | pH | Equilibrium Time (min) | | |
| CR | Rubber seeds + NaOH | 1073 | - | 360 | - | - | 460 | Langmuir, Freundlich, Dubinin-Radushkevich | - | 6–7 | 120 | - | [152] |
| CR | *Shorea robusta* leaf extract + Agnps | 573 | - | 180 | - | 21 | 20 | Langmuir, Freundlich, Temkin, Dubinin-Radushkevich | Pseudo I-order, Pseudo II-order, Intraparticle diffusion, Elovich | 2–10 | 90 | Electrostatic attraction, Hydrogen bonding | [153] |
| CR | *Shorea robusta* leaf extract | 573 | - | 180 | - | 1 | 2 | | - | 2–10 | 60 | Electrostatic attraction | [153] |
| MO | Food waste + ultrasound + H₂O₂ | 573 | 5 | 420 | - | - | 69% | | - | 7 | 60 | - | [43] |
| CR | *Phoenix dactylifera* leaves + Mn | 673 | - | - | - | - | 120 | Langmuir, Freundlich | Pseudo I-order, Pseudo II-order | 2.5, 4.5, 5.8, 8.2, 11.2 | 120 | Electrostatic interaction | [125] |
| Reactive yellow (RY) | *Marine Chlorella* + ultrasound | 723–923 | 10 | 60 | - | 350 | 50 | Langmuir, Freundlich, Temkin | Pseudo I-order, Pseudo II-order | 2.0–10.0 | 2 | Electrostatic interaction, Electrostatic repulsion | [154] |
| Reactive blue (RB 19)/Acid orange (AO) II/Direct red | Sludge-rice husk composite | 773 | 7 | 120 | 0.058 | 30 | 39, 42, 60 | Langmuir, Freundlich, Temkin, Dubinin-Radushkevich | Pseudo I-order, Pseudo II-order, Intraparticle diffusion, Elovich | 7 | 1440 | - | [155] |
| CR | Arjuna (*Terminalia Arjuna*) seeds | - | - | - | - | 170 | 92 ± 5% | | - | - | - | - | [156] |
| CR | Cotton stalks | 673 | 8 | 90 | - | - | 560 | Langmuir, Freundlich, Temkin, Dubinin-Radushkevich | Pseudo I-order, Pseudo II-order, Intraparticle diffusion | 2–10 | 180 | Electrostatic attraction | [126] |

**Table 4.** *Cont.*

| Dye | Biochar Modified Feedstock | Pyrolysis Conditions | | | Pore Volume ($cm^3/g$) | BET Surface Area ($m^2/g$) | Adsorption Capacity (mg/g) or Dye Removal (%) | Isotherm Type | Kinetic Model | Adsorbent Parameters | | Mechanism | Reference |
|---|---|---|---|---|---|---|---|---|---|---|---|---|---|
| | | Temperature (K) | Heating Rate (K/min) | Time (min) | | | | | | pH | Equilibrium Time (min) | | |
| CR | Orange peel + $CO_2$ + steam | 973 | - | 10 | - | 305 | 140 | Freundlich | Pseudo II-order | 2–3 | 1440 | Electrostatic interaction | [123] |
| CR | Litchi peel + hydro-thermal | 1123 | | 60 | 0.588 | 1005 | 400 | Freundlich | Elovich | 4 | 720 | Hydrogen bonding, π-π interactions, Pore-filling, Electrostatic interaction | [150] |
| CR | *Spirulina*/alginate/paper | 723 | 20 | 120 | - | - | 40 | Langmuir, Freundlich, Temkin, Dubinin-Radushkevich | Pseudo I-order, Pseudo II-order, Intraparticle diffusion | 6–8 | 0–120 | Electrostatic attraction, Hydrogen bonding, π-π, | [157] |
| Acid chrome blue/MO | Pine nutshell | 973 | 10 | 120 | - | - | 30, 10 | Langmuir, Freundlich | Pseudo I-order, Pseudo II-order | 3 | 1200 | Electrostatic interaction, π-π interactions | [158] |

Biochars made from sludge have been extensively studied [159]. The Langmuir capacity of the anionic Remazol BB R dye is 125 mg/g [160]. A biochar made out of rice husk-sludge composite prepared at 773 K removed AO II, and the two additional anionic dyes, DR 4BS and RB 19, from solution exhibited adsorption potentials of 42, 60 and 39 mg/g successfully [155].

The separation of 90% of CR from the dye solution was achieved by utilizing a hybrid technique devised by using Arjuna seeds + microorganisms and the ozonation process [156]. Cotton stalk waste is copious around the world. The biochar from cotton waste has been employed to eliminate CR with a 250 mg/g capacity. The addition of ZnO nanoparticles to this biochar increased its adsorption potential to 555 mg/g [126]. The CR dye adsorption capacity of biochar made from orange peel was about 15 mg/g, nonetheless, two tailored biochars made by using steam and $CO_2$ exhibited capacities of 135 and 90 mg/g, respectively. A one-hour hydrothermal carbonisation of litchi skin at 1073 K produced an altered biochar with a surface area of 1005 m$^2$/g and a good CR adsorption potential of 400 mg/g [156].

Spirulina, paper, and seaweed alginate have been combined to create a modified biochar [157]. The goal of this research was to see if algae biorefinery waste and wastepaper could be used to make cost-effective and ecologically friendly xerogels for CR elimination. At the ideal pH value of 6–8, the produced biosorbents possessed a light and porous network structure and a quick dye uptake. The adsorption potential of CR was 40 mg/g.

Biochar was made from raw pine nut husks using fast pyrolysis at 973 K for 3 h [158]. Acid chrome blue K dye sorbates and anionic MO were used to examine the dye adsorption capabilities of the biochar. Before being exposed to a final $FeCl_3$ alteration, the unaltered biochar was treated with modified magnetic biochar and cetyl trimethyl ammonium bromide. Acid chrome blue K and MO have adsorption potentials of 16 and 1 mg/g, respectively, on unaltered biochar. Following the modification, the capacities for acid chrome blue K and MO were 25 and 10 mg/g, respectively.

## 6. Conclusions

This review study discussed the use of biochars and modified biochars for dye removal from effluents. There are several hundred studies on this subject in the literature, and some of them have been cited and detailed in this review. Chemical species, chemical groups, and dyestuff qualities were discussed in this article. A quick overview of treatment technologies followed by a full discussion of the advantages of employing adsorption technology was also covered in this review. The review also highlighted the main features and applications of regular and modified (altered) biochars for color removal.

Certain performance trends can generally be observed and correlated with biochar properties, for example, when using the same raw material source as the fuel source:

➢ The dye adsorption potential/capacity is highly linked to surface area, therefore, the greater the pyrolysis temperature, the greater the dye adsorption capacity;
➢ The biochar yield decreases as the temperature rises. However, this is only true up to roughly 1073 K;
➢ As the micropore and small mesopore walls burn away at temperatures beyond 1073–1123 K, pore volume increases, resulting in fewer but larger pores;
➢ It is important to consider the reaction conditions of temperature, time, and heating rate based upon the type of dyestuff as well as the pyrolysis temperature. The thickness of the pores is also affected by both the type of raw materials and the pyrolysis temperature—dye molecules vary enormously in size, and even small dye molecules are relatively large in comparison to many chemical molecules;
➢ Dye is strongly attracted to oppositely charged sites, so the nature of surface sites on the biochar, depending on raw material and temperature, is extremely important;
➢ Slow pyrolysis yields the best biochar and has the best property control because it produces more biochar, guarantees better pore development, and has a narrower spectrum of pore size distribution.

There are so many different types of modified biochars that it is impossible to list them all; instead, a few brief examples are provided:

➢ Acid or alkali chemical treatment produces biochars with negative and positive surface sites or groups; at temperatures above 823 K, these biochars are known as activated carbons;

➢ Before pyrolysis treatment, sulfur doping of the feedstocks generates biochars carbons with a decent affinity for hazardous heavy metal ions;

➢ Iron oxide-doped modified chars have demonstrated to be particularly attractive adsorbents for both anions and metal cations, as well as chromate, using ion exchange;

➢ Coating the biochars has been exceedingly successful; for instance, coating with chitosan provided adsorption capabilities of 5-fold and 20-fold *w/w*, placing these altered biochars in the super-adsorbent category.

The future of biochar technology appears bright overall. Several biochars are made from waste biomass resources and are therefore carbon neutral, making them a cost-effective and environmentally friendly product with many potential applications. The cited literature contains numerous gaps that will be essential to address for designing treatment plants for dyehouse effluent biochar. However, recent articles have addressed the "tomorrow's path" which should be followed in future adsorption studies.

**Author Contributions:** Original Draft, Writing—Review and Editing, P.P.; Original Draft, S.S. and J.S.; Original Draft, Writing—Review and Editing, M.A.; Funding acquisition, Conceptualization, Resources, Project administration, Supervision, Writing—Review and Editing, G.M. All authors have read and agreed to the published version of the manuscript.

**Funding:** The authors would like to thank Qatar National Research Fund for their support of this research through an award NPRP-11S-0117-180328, the Supreme Committee for Delivery and Legacy (SCDL), and to Hamad Bin Khalifa University, Qatar Foundation, for an award to H.F. Any opinions, findings, and conclusions, or recommendations expressed in this material are those of the author(s) and do not necessarily reflect the views of HBKU or QF or SCDL.

**Institutional Review Board Statement:** Not applicable.

**Informed Consent Statement:** Not applicable.

**Data Availability Statement:** Not applicable.

**Acknowledgments:** The authors wish to thank Hamad Bin Khalifa University (HBKU), Qatar Foundation (QF), and Qatar University for their patronage.

**Conflicts of Interest:** The authors declare no conflict of interest.

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
