# Peer review of "A Review of the Removal of Dyestuffs from Effluents onto Biochar"

_separations, doi:10.3390/separations9060139_

Round 1
Reviewer 1 Report
In this review, the use of biochars and modified biochars for dye removal were discussed. The manuscript is suitable for publication only after major revision.
- Why is adsorption suitable for the treatment of dyes? What are the advantages over photocatalytic degradation?
- What is the experimental mechanism of biochar treating dyes? Is it chemical adsorption or electrostatic action?
- It is suggested that the author briefly explain what enlightenment this review has for other researchers.
- The references selected by the author are not new enough. Please add some latest research work.
- Adsorption is a commonly used water treatment technology, and the author should introduce commonly used adsorbents. Several relative papers are suggested to be cited. (10.1016/j.jhazmat.2021.128062, 10.1021/acsami.1c22035, 10.1016/j.jhazmat.2020.123810, 10.1016/j.clay.2018.12.017).
- There are some grammatical errors in the manuscript, and please revise and submit it again.
Reviewer 2 Report
This review article comprehensively provides an overview on the employment of biochar for dye removal. The article is well written and contains useful information for the readers. In particular, I appreciated the detailed discussion of the different types of dyes and their possible adsorption interaction with biochar, as well as the rationalization of the biochar properties-adsorption performances correlation summarized in the conclusions. For these reasons, the manuscript can be considered for publication after some minor revisions. See details here below.
- (Figure 1) The quality of figure 1A is bad. Replace this image with a higher resolution picture if possible.
- (Figure 3) Please replace “HCL” with “HCl”.
- (Lines 280-281) Please replace “book” and “chapter” with “review” or "article".
- (Line 405) “ii.” is repeated two times.
- (Lines 435-436) “The temperature and type of feedstock used will determine the properties.”. This part of the text should be greatly expanded prior to discuss the adsorption of dyes onto biochar in detail. Such discussion may provide to the readers basic information on how biochar properties (porosity, specific surface area, surface functional groups, etc.) can be affected by pyrolysis conditions and biomass feedstock. Please refer to previous review articles (DOI: 10.1007/s11157-020-09523-3).
- (Lines 363-367) It should be mentioned the employment of biochar as support (DOI: 10.1002/jctb.6279, DOI: 10.1039/C8RA02258E) and, more recently, as co-catalytic nanoparticles/dots (DOI: 10.3390/catal11091048, 10.1016/j.cplett.2020.137428) to enhance adsorption and catalytic properties of TiO2-based photocatalyst aiming at improving dyes removal performances by enabling synergistic adsorption and photodegradation. Please add this information in the manuscript with relevant references as suggested.
- (Tables 1-4) Detail on the type of dye can be removed in the Table as the information cationic or anionic is already present in the caption.
- (Table 3 and Table 4) The captions must be revised. Table 3 includes cationic and not anionic dyes, and vice versa for Table 4.
Round 2
Reviewer 1 Report
The authors have addressed the critical issues in the revised manuscript. it is now suitable for publication.